# Community voices: Exploring beliefs, attitudes, practices and recommendations for improving stroke prevention and stroke care in rural and urban communities in Nigeria

Ezinne O. Uvere[1,2], Carolyn Jenkins[3], Ezinne S. Melikam[4], Oyedunni S. Arulogun[2], Omotolani T. Ajala[1], Adewale E. Ayodele[1], Olayinka J. Adebajo[1], Osimhiarherhuo Adeleye[1], Olumayowa Ogunronbi[1], Chukwuemeka Nwimo[1], Michelle Nichols[5], Oluwafemi A. Popoola[6], Joshua O. Akinyemi[7], Raelle Tagge[8], Rufus O. Akinyemi[9,10], Onoja M. Akpa[7], Ayodeji Salami[11], Olalekan J. Taiwo[12], Ayodeji Adebayo[6], Paul Olowoyo[13], Temitope Ilori[6], Olanrewaju Olaniyan[14], Richard W. Walker[15], Bruce Ovbiagele[8‡]*, Mayowa O. Owolabi[1,10,16,17,18‡]*

1 Department of Medicine, College of Medicine, University of Ibadan, Ibadan, Oyo State, Nigeria, 2 Department of Health Promotion and Education, College of Medicine, University of Ibadan, Ibadan, Oyo State, Nigeria, 3 College of Nursing, Medical University of South Carolina, Charleston, South Carolina, United States of America, 4 Department of Public Health, Clemson University, Clemson, South Carolina, United States of America, 5 Department of Health Sciences and Research, College of Health Professions, Medical University of South Carolina, Charleston, South Carolina, United States of America, 6 Department of Community Medicine, College of Medicine, University of Ibadan, Ibadan, Oyo State, Nigeria, 7 Department of Epidemiology and Medical Statistics, College of Medicine, University of Ibadan, Ibadan, Oyo State, Nigeria, 8 Weill Institute for Neurosciences, University of California, San Francisco, San Francisco, California, United States of America, 9 Neuroscience and Ageing Research Unit, IAMRAT, College of Medicine, University of Ibadan, Ibadan, Oyo State, Nigeria, 10 Center for Genomic and Precision Medicine, College of Medicine, University of Ibadan, Ibadan, Oyo State, Nigeria, 11 Department of Pathology, College of Medicine, University of Ibadan, Ibadan, Oyo State, Nigeria, 12 Department of Geography, Faculty of the Social Sciences, University of Ibadan, Ibadan, Oyo State, Nigeria, 13 Department of Medicine, Afe Babalola University, Ado Ekiti, Nigeria, 14 Department of Economics, Faculty of Social Sciences, University of Ibadan, Ibadan, Oyo State, Nigeria, 15 Population Health Sciences Institute, Newcastle University, Newcastle upon Tyne, United Kingdom, 16 University College Hospital and University of Ibadan, Ibadan, Oyo State, Nigeria, 17 Lebanese American University of Beirut, Beirut, Lebanon, 18 Blossom Specialist Medical Centre, Ibadan, Nigeria

‡ These authors are joint senior authors on this work.
* mayowaowolabi@yahoo.com (MOO); Bruce.Ovbiagele@va.gov (BO)

## Abstract

### Introduction

Globally, stroke is a leading cause of mortality with higher incidence in low- and middle-income countries. However, stroke beliefs and knowledge among community partners are essential considerations for tailoring of interventions in West Africa.

### Purpose

To describe differences in beliefs, attitudes, and practices related to stroke risks, prevention, and care delivery from alternative/complementary providers/healers, orthodox/modern medicine/health care providers, community members and leaders in Nigeria.

---

**Data availability statement:** Per the conditions of the informed consent document, the study team destroyed audio recordings after verifying the accuracy of the transcripts. Also, the informed consent signed by participants stated that the study team would employ all means possible to maintain the confidentiality of their personal information and that data collected would be used solely for the purposes of this study, with access limited to authorized study personnel only.

**Funding:** The study is supported by the National Institutes of Health (NIH) grant: ARISES (R01NS115944-01). The funders of this study had no role in study design, data collection and analysis, decision to publish, or preparation of the manuscript.

**Competing interests:** All authors declare no competing interests.

## Methods

Six focus groups with community members and leaders (n = 57) and key informant interviews with health providers (n = 24) from alternative/complementary and orthodox/modern medicine providers were conducted to qualitatively explore beliefs, attitudes, practices, and recommendations related to stroke in urban (Ibadan) and rural (Ibarapa) communities in Nigeria. The Socio-Ecological Model guided selection of participants, and the Health Belief Model guided the development of questions for participants.

## Results

Participants perceived stroke as disabling, though manageable, and having higher odds of repeat stroke for survivors. High blood pressure, stress, sleep issues, heredity, and lifestyle factors were some stroke risk factors perceived by participants from both sites although God, witchcraft/evil people were reported by rural participants. Hospital visits and consumption of herbal concoction, self-medication and visit to church for prayers were some actions taken to manage stroke by both urban and rural participants. Low literacy levels, limited funds, fear of and distance to hospitals, and absence of insurance were some barriers to uptake of recommendations from orthodox medicine practitioners which are drivers to unorthodox practitioners. To improve stroke care and prevention across communities, free risk factor screening, indigenous stroke awareness programs via print, audio-visual and electronic media were suggested by all participants.

## Conclusion

Diverse beliefs and practices are related to stroke risk factors, prevention and care and barriers with obtaining care. There is need to work across systems to improve stroke prevention and care in communities.

## Introduction

Stroke is one of the topmost determinants of death and disability worldwide [1,2], with increasing rates recorded across Africa [3–7]. In most high-income countries (HIC), stroke rates and outcomes, such as mortalities and impairments, are declining while rates among adults in Africa continue to rise [8–13]. There are several approaches towards addressing stroke related deaths and disabilities [14,15]. However, multiple and interrelated factors pose great challenges to access preventive services and adoption of evidence-based recommendations and therapeutic regimens towards mitigating the growing incidence of stroke [16]. Structures identified include demographic, personal, socio-economic, and traditional factors as impacting wellbeing [15]. Conceptual models have been utilized to understand the multi-dimensionality of human behavior and action. The Stroke Investigative and Research Education

Network (SIREN) utilized the Health Belief Model (HBM), as well as some elements of Arthur Kleinman's Explanatory Model of Illness [17] to unravel knowledge, beliefs, attitudes, and practices towards stroke prevention and care. Findings were organized by groups using the Social Ecological Model (SEM) [2,18,19].

Given the wide variation in sociocultural paradigms across contexts, the employment of qualitative approaches toward stroke-related health behaviors is widely recommended. Underscoring beliefs relating to stroke from the lens of underrepresented communities are essential to the design of prevention programs that are locally relevant [20]. Hence, this paper presents key informant interview (KII) and focus group (FGD) findings that explored stroke risks, susceptibility and severity, benefits, barriers, and cues to action for decreasing stroke risks as well as improving stroke outcomes for individuals, families, organizations, communities, and societies in Nigeria and beyond. Data were collected across an urban (Ibadan) and a rural (Ibarapa) community in Nigeria. These community voices help to: 1) lay the groundwork for exploring community members' attitudes, beliefs and practices related to stroke; 2) identify methods and messages for improving stroke prevention and care; and 3) improve participation in the new Stroke Information and Surveillance System (SISS) that is being tested in the two communities.

## Methods

Data collection instruments, analyses, and protocols were developed and tested prior to engaging the community members in ARISES. We followed the standard processes of reporting and best practices for qualitative research as identified by both the 32-item Consolidated Criteria for Reporting Qualitative Research (COREQ) checklist [21], and the Standards for Reporting Qualitative Research: a Synthesis of Best Practices [22]. These guided our processes and the development, testing, implementation and reporting of the findings [2]. COREQ guidelines for presenting the research processes includes three domains: research team and reflexivity, study design, followed by analysis and findings [21]. Results are presented by using two guiding theories- the Health Belief Model (HBM) [23] for questions and answers and the Socio-Ecological Model (SEM) [24] as the framework for organizing the participants answers by groups.

### Domain 1: Reflexivity and bias in ARISES research team

Reflexivity helps to capture relationships and influence of the researchers and the participants [25,26] and the role of subjectivity of the team in the research process [25,26]. The overall goal of ARISES study is to deploy and validate a first-of-its-kind scalable integrated mHealth community-based interactive Stroke Information and Surveillance System (SISS) for reliable measurement and real-time tracking of the population's burden of stroke while simultaneously building sustainable capacity for improving stroke literacy, early presentation and outcome in two pre-existing demographic surveillance sites: one urban and one rural area in Nigeria" [27]. The team is diverse, as members have prior research experience [2] related to their research activities in ARISES. The team is led by multiple Principal Investigators (PI) (MOO and BO) who are both expert researchers and neurologists that provide clinical care to persons with stroke and lead numerous research studies focused on stroke prevention and care in Nigeria and beyond. The team for community engagement includes investigators and staff with expertise in community engagement and both qualitative and quantitative research. The qualitative components of ARISES are led by CJ and OSA and EOU; all with expertise in community engagement and qualitative research, particularly in under-resourced communities in both Nigeria (OSA, EOU & CJ) and the African American population in the Southern United States (CJ). Other team members involved in the data collection for qualitative research (FGD and KII) have prior experiences in research data collection in Nigeria. All have completed some form of tertiary education, were either field/community managers/supervisors, public health professionals, and each is knowledgeable about the culture of the communities.

The teams communicate via Zoom at least two to three times per month. The team (CJ, OSA, EOU, ESM) co-developed the ARISES community engagement study protocol; co-designed the FGDs and KIIs guides; organized workshop sessions on FGD and KII methodologies; scrutinized and reviewed findings from the KIIs and FGDs with the

team; worked collaboratively to analyze data; provided oversight to the data translation process; reviewed the reflexivity and biases; developed and proofread this manuscript [2]. Other team members then received a copy of the qualitative research processes and participated in the workshop sessions; field tested and reworded the FGD and KII questions; implemented the FGDs and KIIs. Data collection was conducted by selected indigenous field staff. These field staff have experience with stroke research activities with additional training in public health, epidemiology, education, and development. They performed the role of either (moderators, note takers) at each site, and had some prior relationship with some of the study participants (See acknowledgment section) [2]. Despite having had some skills in conducting qualitative research, participation in this study by the FGD data collection team was guided by the study's protocol to maintain similarity of procedures and methodologies.

## Domain 2: Study setting, design, analysis and findings

The study settings have been detailed in previous publications [27], particularly the urban community that participated in the SIREN study [2]. The settings for the FGDs varied but most were in community sites (including neighborhood centers, worship surroundings, town halls), while the KIIs were planned/conducted in the providers' offices and hospitals or workplaces. Only the leader and note taker were present with the participants during each of the FGDs or KIIs.

The rationale for using oral KIIs and FGDs was to capture the attitudes, beliefs, and practices related to stroke prevention and care practices that may not have been captured in prior surveys and qualitative studies. Each of the two sites planned twelve KIIs (or until saturation of data) with both unorthodox medicine practitioners, and Orthodox medicine practitioners that served each community and twelve FGD sessions with community members and leaders to discuss stroke prevention and care in each of the two communities. To standardize the data collection processes, relevant resources including forms for informed consents, data collection and note-taking, demographics, and data analyses were developed. The methodology adopted for the FGD and KII data collection sessions was guided by Kruger and Casey [28] while Tremblay [29] influenced the in-depth interview process. The study obtained approval from the Ethics Review Committee of the University of Ibadan/University College Hospital (UI/EC/19/0629). Written informed consent was mandatory for all participants before commencement of any form of activity, [Ethical approval document attached]. Recruitment of participants for this study commenced on the sixth of September 2021 and ended on the ninth of November 2021.

## Theoretical models

Theoretical frameworks that guided our FGD and KII questions, processes and analyses are the Health Belief Model (HBM) and the Social Ecological Model (SEM) [23,24,30].

The Health Belief Model [23,30] focuses on understanding an individual's perspectives to help modify behaviors and includes perceived susceptibility (*a person's assessment of his/her likelihood of developing a particular health problem or condition*), perceived severity (probability that a person will change their health behaviors to avoid a consequence depending on how serious they believe the consequences will be),perceived benefits (*how the effectiveness of various available actions to reduce the risk of illness are perceived*), perceived barriers *(a person's assessment of the obstacles that may prevent them from engaging in a recommended health action*), cues to action (*events – whether internal or external that trigger a person to consider or take action related to their health*), and self-efficacy (*a person's perception of his/her ability to successfully perform a behavior*).

The Socioecological Model (SEM) [31,32] is a multilevel conceptualization of health activities adapted from US Centers for Disease Control and Prevention that includes:

• intrapersonal/individual – knowledge, attitudes, beliefs, skills, and behaviors

• interpersonal – individual relationships with family, friends, social networks, support groups and cultural context

- institutional/organizational – organizations, social institutions (schools, health care/administration, business, faith-based organizations), as well as rules, regulations, policies, and informational structures

- environmental/community – relationships among organizations, institutions and environments

- structures, policies, systems--local, state, national, international policy, laws & actions and their built environments.

**Sampling and sample recruitment**

Participants for this study were unorthodox medicine practitioners, orthodox medicine practitioners (Medical Doctors [MDs], Physician Assistants [PAs], Neuro-professionals, Nurse Practitioners [NPs], Registered Nurses [RNs]), as well as community leaders and members. Community engagement coordinators recruited participants across rural and urban communities through purposive, convenience, and snowball sampling. Their invitation to participate in either the KII or FGD was done by the ARISES staff, largely through face-to-face contact and telephone calls or a combination of both. Community leaders and members (n = 57) participated in one of six FGDs, and community-based health care providers (n = 24) both from unorthodox medicine practitioners, and orthodox medicine practitioners (physicians, physician assistants, nurse practitioners, and registered nurses) participated in KIIs across the two communities in Nigeria. More detailed characteristics of the FGD and KII participants have been described under the results section below. All KII participants contacted and recruited participated in the study. Only five [5] people from the FGD sample refused to participate in the FGD session due to conflicts of time.

**Data collection**

Interview guides for both the KIIs and FGDs were used for data collection. These were collectively reviewed/refined with all investigators and field staff through teleconference meetings on Zoom. An overview of the key questions is shown in the supplementary section. The staff participated in online training and practice prior to beginning recruitment and interviews. The site coordinator (EOU) and staff reviewed, edited, and worked with community members to translate and back translate into the local language/dialect as needed. All data were digitally recorded; field notes were made during the interview; and the digital recordings were transcribed verbatim. The process developed by Beaton and colleagues [33] provided a guide for translation of both the questions into the language or dialect as well as translation of the results into English. Following translation into English for those FGDs and KIIs administered in another language, review and editing of all transcripts at each site, the digital recordings, verbatim translations, and field notes were loaded into REDCap [34], a secure data management and storage system. Personal identifiers were removed from the transcripts that were used for analyses. Four [4] of the FGD and KII transcripts were returned to a sample of participants for review prior to analysis for confirmation of accuracy. No edits were made by the participants as they acknowledged that transcripts were true representation of their opinions.

**Domain 3: Analysis and findings**

**Analysis.** The guiding methodological approach selected for identifying, analyzing, and presenting the data is thematic analysis [35] using the structures from the HBM [23] and SEM [32,36] for generating the initial analytic approach and codes. Thematic analysis, as an independent qualitative descriptive approach, is mainly described as "a method for identifying, analyzing and reporting patterns (themes) within data [28,37,38]." The data findings were coded by one of the community engagement co-investigators (CJ) and (ESM) who had previously worked with the team at University of Ibadan while the project director (EOU) collated and synthesized demographic data as well as qualitative data and finalized and submitted the manuscript. All authors and the PIs (MOO and BO) met at least twice a month with staff and investigators, guided the team and reviewed activities and progress. Major themes or nodes

were first identified using the domains from the HBM and SEM [17,32] and were reviewed through a continuous process of data segment comparison based on the qualitative research techniques described by Huberman and Miles [39] and Patton [40]. A codebook was developed defining themes, and a numeric theme code was assigned to each category of text responses. Participants' responses were coded and sorted accordingly into differing categories. Microsoft Word was used to create tables sorted on theme code. An iterative process of reading and rereading the data was used to refine the categories and ensure the coded responses fit well into the categories. The analysis was systematic and involved categorizing data into the two theoretical frameworks [23,36]. Additionally, NVivo 10 software [41] was used to re-analyze the data first by using the HBM [23] and each of the coded data segments was then coded into the appropriate level of the SEM [19]. The coded data segments from the first coding and then from NVivo coding were extracted, compared and differences resolved, condensed, and summarized with examples of major and minor themes (including diverse themes) [39,42,43]. The themes and quotes were then provided for feedback from each ARISES team member and there were no disagreements when reviewed by the authors. The summary of steps for the methods and coding processes (See Fig 1 below) was adapted from a previous study (SIREN) [2] and includes the major and minor or divergent themes which are presented using a condensed framework integrating the variables in the HBM [30] and the SEM [24,32].

## Results

### Overview of participants and data findings from unorthodox and orthodox health providers, and community members and leaders

**Description of participants.** A total of one hundred and fourteen persons (114) across the communities in Ibadan (urban) and Ibarapa (rural) participated in one of twelve FGDs and twenty-four health care providers (both unorthodox medicine n = 6/site and orthodox practitioner providers = 6/site) participated in the KII respectively. The KIIs lasted for 60 minutes or less while FGDs lasted 60–90 minutes.

Twenty-four [24] participants (13 females and 11 males) were involved in the KII session and within the range of 39–69 years [mean age: 50.9 years]. About 50% of these participants were resident at either the urban or rural sites respectively. Of the 24 participants, 22 participants were married [with one never married and one widowed participant] and from Yoruba ethnic affiliation. Only one participant was from the Igbo ethnic group. With respect to their educational attainment, about 12.5% have obtained a primary school certificate; 21% a secondary school certificate, 50% a university degree while 12.5% have obtained a postgraduate degree. Only 4% had no form of education. In terms of their roles at their various communities, an equal proportion (50%) of participants at both the rural and urban sites were either orthodox (Matron/Nurses, Community health officers, medical doctors, physiotherapist) or unorthodox medicine (Patent Medicine Vendor, Bone setter, Masseur, Traditional healer, Herbal medicine practitioner) practitioners.

A total of one hundred and fourteen persons (114) across the communities in Ibadan (urban) and Ibarapa (rural) participated in one of twelve FGDs. Female participation was lower than male participation across both FGDs (Females – 39; Males – 75). All FGD participants across both study sites (n = 57) were Yoruba. Of this number, 52 participants from the rural sites were married [with 3 single, 1 divorced or widowed respectively] while 51 participants from the urban site were married [with 5 widowed and 1 single] at the urban site. At the rural site, participants were within the range of 19–98 years. Male participation was greater [39] compared to females [18] with mean age of males at (54.8 years) while females had a mean age of 46.6 years. At the urban site, participants were within the range of 35–90 years of age. Similarly, males were more (n = 36) compared to the female (n = 21) participants. The educational profile of all participants revealed that majority of the participants have either obtained a primary, secondary or tertiary level of education. Only few participants have no education. In terms of their occupation across the communities, about 24% of the FGD participants (community members) at both the rural and urban sites were involved in trading; 18% were civil servants, 11% were religious leaders, 17% were retired, 21% artisans, with 5% being traditional medicine practitioners.

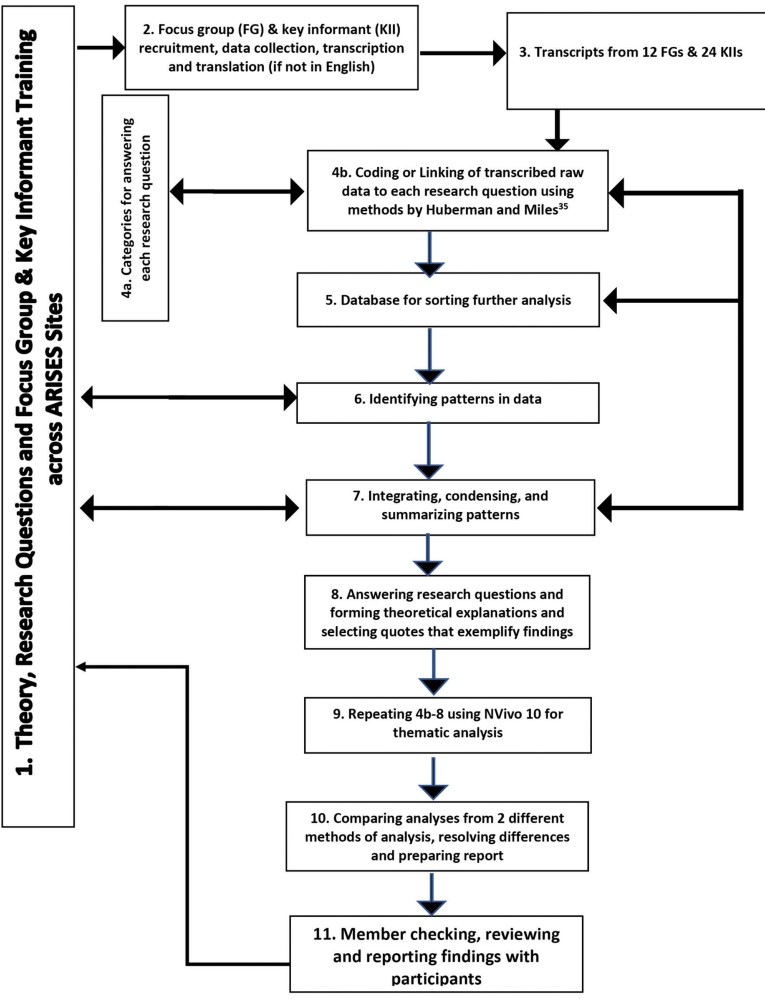

**Fig 1. Overview of steps in the ARISES qualitative process for reporting, analyzing and reporting findings [42].**

**Findings from the participants in KIIs and FGDs.** S1 and S2 Tables (under the supplementary section) show the questions that were developed to help ARISES obtain data from the participants (as described above). Data presented in S1 Table summarizes the answers to the questions for health providers (both unorthodox and orthodox health care providers) and data presented in S2 Table summarizes FGD data from community leaders and members. For both tables, the major themes are listed first and followed by other less common themes using a condensed framework integrating the Health Belief Model [30] with the Socioecological Model (SEM) [19]. Findings from both sessions are presented below with direct quotes of the participants in quotation marks.

**Findings From the Key Informant Interviews (KII) with orthodox and unorthodox practitioners across study sites**

1. *Risk susceptibility*: The dominant themes for causes of stroke were somewhat similar across both areas except with few variations. At the rural areas among both orthodox and unorthodox providers, causes centered around uncontrolled blood pressure, God, evil people or witchcraft, unhealthy diets, use of alcohol while others reported bad air, lack or stressful exercise, smoking, older age, envy, drugs, lipids, snuff, stress, sugar diabetes or bad water. At the urban

sites, causes revolved around stress/strenuous jobs and heredity and/or blood pressure. Other identified reasons included diseases (heart, kidney); lifestyle/sedentary lifestyle; bad air, alcohol, tobacco, evil people, herbs from alternative medicine providers, refusing treatment, obesity, older age, sin, state of the country. At the rural sites, unorthodox and orthodox care providers alluded to understanding these risk factors and are taking steps to prevent them. However, there were still some perceptions of the likelihood of stroke event in every person regardless of adopting prevention practices. "*Yes, there is nobody that cannot be affected with this sickness of stroke.*" – **(Traditional Bone Setter – Ibarapa).** Further, the most common environmental risk factors for stroke across sites by both orthodox and unorthodox providers were *heredity* and/or *high blood pressure* as stated below "*Those that are obese, those with hypertension, high blood pressure without monitoring or treating it, diabetes, obesity; people that take alcohol, smokers, stressful jobs and do not take enough rest or concerned about their health.*" (**Matron – Ibarapa**). At the urban site, for orthodox providers, although they are in terms with the risk factors, some indicated their likely risk for stroke due to external stressors including increased workload, reduced mobility/physical exercise. "*Well, I am at risk. The workload is very cumbersome. I think one of those things that causes high blood pressure is inability to have enough rest….and I will be restrained to a place… staying in one place, not been able to use some part of the body.*" – (**Chief Nursing Officer – Ibadan**). Very few unorthodox providers denied likely risk because of their use and access to traditional treatment options as stated below: "*Never, I can't have it* (stroke) *because of traditional medicine…. There are things that can be used to prevent it. We also believe that herbs does wonder with the grace of God.*" – **(Traditional Bone Setter – Ibadan).** Regarding what people fear about stroke*, m*ost practitioners mentioned paralysis, unable to get up or walk, confined to spot, incapacitated most of the time, inability to eat or go to toilet by oneself as stated in the statements below: "*What I fear most about the ailment it is that it does not allow the patient and the family to have rest. It is an ailment that takes most of the relatives' time and money consuming…Even most time, the patient does not get healed and eventually dies*". **(Traditional bone setter 2 – Ibarapa)**. At both sites, orthodox and unorthodox providers stated that persons given to smoking, alcohol intake, who are obese; are either children, men, elderly with other disease; those who skip breakfast and do not check their blood pressure regularly are susceptible to stroke "*Firstly, anyone who is experiencing dizziness from within or sudden blurring of eyes… such person will develop stroke… Secondly, when someone fails to take breakfast … this can cause deficiency in vision and a platform for developing stroke. You see, drinking of alcohol, … Taking Indian hemp (hard drugs) with not eating on time or not eating at all, may cause stroke, as there will be shortage in blood in the body* (Traditional healer – Ibarapa).

2. ***Risk Severity***: Per the physiological causes of stroke, orthodox providers at both sites highlighted disruption/rupture of blood supply to the brain accompanied with deficits in speech, mobility. In contrast, unorthodox providers in the rural site were only able to mention that stroke occurs suddenly with no symptom of any accompanying sickness as stated below: "*There is nothing that happens to people that develop stroke, it befalls them suddenly while seated. Suddenly, they just fall. There is usually no symptoms of sickness…Stroke patients don't have any pre-stroke sickness or ailment*". **(Traditional healer – Ibarapa).** Also, providers were able to identify some of the known signs of stroke including high blood pressure, inability to sleep, feelings of numbness, weakness. However, malaria fever was indicated by an unorthodox provider in an urban site as stated here: **"***The fingers and toes can try to get hooked at times without control. Another sign is inability to sleep. These signs gradually point towards stroke. The third sign is heavy headache. The fourth sign that I can add is malaria fever and its lack of treatment in the person*" (**Patent Medicine Vendor – Ibadan**). In terms of impact, stroke is a burden to affected families as it reduces the usefulness of and increases the sufferer's economic dependence on others. This was the position of both orthodox and unorthodox practitioners as substantiated: "*Somebody who has a stroke is a problem to the society, to the family; then it constitutes a lot of problem to that person because it will affect him financially, that person will not be able to go to work or do all the activities he has been doing before; it affects the place of work.*" (**Chief Nursing Officer 1 - Ibadan**). While describing the duration of stroke event and the degree of severity, generally all providers alluded to the short-lived

duration of stroke which is usually exacerbated by the absence of affectionate care by a loved one. They were also of the opinion that some stroke sufferers can live long depending on access to quality care: *"Somebody who is an elite or has someone taking care of him, they can live for 2–5 years but if there is nobody to care for them, they give up within 1–2 years" **(Masseur – Ibarapa)**.* Overall, all practitioners believe that stroke sufferers should receive medical treatment which should incorporate administration of multi-faceted therapies/specialties including physiotherapy, health educators, speech therapists etc. *"Exercise of the part affected, regular medication, diet monitoring as this important" **(Masseur – Ibarapa).*** However, unorthodox practitioners mentioned the need to include both orthodox and unorthodox medical treatment during stroke management. Additionally, they maintained that early presentation to these facilities is key for an improved treatment outcome. *"It could be either modern medicine or traditional medicine. But the most important is early presentation of the ailment. Both treatment is good. The kind of recommendation that I can outline include going to hospital regularly. The reason why I said hospital is they will quickly know if your health status is good or not. Early presentation by the patient is the key. When they go early to the to the hospital, the doctor will know whether there are other disease within the system of patient". **(Traditional bone setter 2 – Ibarapa).***

3. ***Benefits to action*:** Despite suggestions that stroke sufferers should receive medical treatment in the event of stroke, practitioners noted that unorthodox facilities (such as herbal homes, religious worship centres) are usually consulted largely by community members in the event of stroke. This action was opted for based on prior experience and testimonials from friends and neighbours. Further, community members opted for this action due to an underlying perception of stroke as a spiritual attack that needs a spiritual approach for management. *"Some people believe that they are being inflicted by the evil one or an enemy and for this reason some believe that going to church for deliverance … Some believe in church and so would prefer to take them to church for such an occurrence. … some do not like to be carried about but would prefer to be left in the house. Some would even say they do not want to go to the hospital … They get scared and they go to a traditional herbalist to bath with soap or herbs."* (***Matron – Ibadan***). Regarding the benefits of the actions taken by communities in the event of stroke, sufferers that visited orthodox facilities promptly noted slow recovery while those that patronized unorthodox facilities had negative outcomes. However, unorthodox practitioners perceived that sufferers that adopted their services had a positive outcome. *"…We went there with herbal concoction; he drank it and everything was flushed out. My joy now is that he did not die….He drank it and he immediately started getting well. If he had gone for surgery and was asked to pay 10million….but the bottom line is that they involved me." **(Traditional healer – Ibadan).*** In terms of adherence to recommended stroke preventive activities, these are usually adopted by the community members. Factors that influence adoption include financial buoyancy, level of education, reputation of healthcare provider/facilitators, experience from other sufferers. However, adherence can be negatively impacted due to low financial status: *"…. Most people don't go to the hospital because they know that they will do tests, buy drugs, and pay bills and they don't have the money. Most of these people can't afford to eat. There's no way they can go to the hospital." **(Patent Medicine Vendor 2 – Ibadan).***

4. ***Barriers to adoption of recommendations:*** Generally, poverty and ignorance is the main barrier to the adoption of recommended actions of healthcare practitioners. This is because the lifestyle recommendations such as changes in diet, physiotherapy and purchase of drugs requires huge sums of money whose costs keeps increasing given rising inflations in the country's economy. Increased cost of orthodox treatment options was identified as one of the major driver to patronizing unorthodox practitioners: *"The first one is money. Once we advise them to take the person to the hospital, they will tell us that it will gulp money as treatment is not free in the hospital. Another problem is that they will say once they take the person to the hospital, they will prescribe many drugs that will be expensive to buy. They might even tell them to buy walking sticks for the patient but if there is no money, they will not do it. That is why they end up saying they should take such a person to the herbalist and see how the treatment goes."* (***Patent Medicine Vendor – Ibadan***). Further, negative perception about hospitals, especially those that provide specialized treatments for stroke

was mentioned as another barrier. This is due to perceived stress that is involved in navigating the various units that provide different services required for stroke management within the hospital. For instance, community members have a perception that referral to a hospital in the event of stroke especially tertiary level healthcare facilities is usually a death sentence for the stroke sufferer. As a result, they prefer to consult unorthodox options: "*People are afraid of hospitals; they don't want to pass through any rigorous action in the hospital. Any stress in the hospital they don't like it…. They want quick things… so they want fast action, they want fast intervention, and it is not possible*" (**Chief Nursing Officer 2 – Ibadan**). "*Some people also believe that when a person with stroke goes to the hospital, he/she will not come back home alive; may spend a long time in the hospital where they get to use them for medical experiments and tests or may not survive it because they believe it an evil infliction and not an ill-health issue.*" (**Matron – Ibadan**).

5. *Cues to action:* Some of the reasons that trigger people to accept a recommended health action towards stroke prevention included seeing someone who have had stroke and recovered, fear of stroke and associated disability and eventual death from stroke events: "*…and when they have examples of those that have had it before and recovered.*" *(Masseur – Ibarapa); "People accept recommended health actions when they see a stroke survivor that got treated in the hospital*" (**Matron – Ibadan**).

6. *Role of Institutions in stroke prevention across communities*: Various institutions, both private and public organizations were mentioned as important in the management of stroke prevention across communities. They could contribute by providing well-equipped facilities for diagnosis and treatment especially in the rural sites. Also, free treatment and some form of subsidy to supplement stroke treatment and management across health facilities were recommended. These strategies, according to the participants, will motivate stroke sufferers and people within their community to make timely visits to these facilities. Further, the need for community sensitization on stroke risk prevention and management in local languages via audio, print, social media was highlighted. This sensitization process should be implemented by involving indigenous gatekeepers across communities:

"*…they should come up with… public lectures, sensitization regularly, counselling regularly. They can also provide fliers in the local language/dialect and also in English. So you collect whichever you will understand. Then they can air on (radio, television) on high blood pressure and stroke…They can also use the social media rolling our numbers by which people can contact them. Also, a toll-free number to also call. They can also engage our service as their representative in our community to help them propagate, with a token as transport fare*" (**Patent Medicine Vendor – Ibarapa**).

7. *Recommendations for improving the SISS approach*: To track the burden of stroke across communities, the ARISES team developed an approach titled Systematic Investigation of Stroke Surveillance (SISS) which is an innovative M-health approach that educates communities on how to identify stroke events and to facilitate prompt reporting of stroke events to the research team. To make the SISS deployment and implementation culturally sensitive, participants suggested that the mobile hotline telephone line component of SISS should be toll-free. Further, awareness of the SISS should be created across sites using print media including pamphlets, stickers and fliers. These resources should contain additional information about the SISS and mobile telephone hotline numbers to facilitate community members timely report of stroke. Additionally, as stroke cases are identified, participants expressed concern regarding place of referral at their communities for their management. Hence, they suggested that facilities should be set up at each site which will serve as a facility where they can refer people who show signs/symptoms relating to or actual stroke to: "*The SISS is good as I believe it will promote prompt recognition of people with stroke, and it will ensure they get quick attention which will generally improve their prognosis*". *(Matron – Ibarapa).*

"*You can have a small post here where your people will stay so that if we see anyone that has stroke, we will take them there.*" (**Patent Medicine Vendor** 2 - **Ibadan**).

**Findings from the Focus Group Discussions (FGD) across sites**

1. *Risk susceptibility*: According to the participants, numerous factors can increase one's risk to stroke. Some of the factors listed by the participants revolved around known factors such as high blood pressure, poor dietary practices, sleep disorders; stress (due to too much thinking and economic difficulties). Participants mentioned risks that are mythical included evil forces, evil wind, the act of sex in a standing posture or the act of pouring water on the head during shower: *"…it (stroke) is used by evil forces to afflict people through the wind."; "High blood pressure", "Too much thinking is enough to cause stroke sickness." "Whoever finds it difficult to sleep regularly is also prone to this sickness"* – *(FGD Participant, Ibadan)*.

*"If a man delights in womanizing, he is prone to stroke." "Yoruba believed that whoever is having sex with his wife in a standing position will have his legs shaking as time goes by. It is possible for him to have stroke.", "…if he showers and pours water on his head, it can cause it (stroke)." (FGD Participant, Ibarapa).*

2. *Risk severity*: Stroke is seen as a disease that distorts normal body function. According to participants, stroke upon its occurrence affects the physiological functioning of its sufferer. It could result in paralysis of the body and disability if not timely managed. Stroke was perceived as a sudden event which happens unannounced and doesn't leave its sufferer the same. It can be identified with the onset of headache, high blood pressure, sleeplessness, inability to move the limbs, poor vision. Upon its occurrence, stroke sufferers are avoided by close friends, neighbors and community members due to the bodily discomfort and unpleasant body discharges that emit from the stroke sufferer. One of the chief problems that a stroke causes for the sufferer and their families revolves around the financial burden it is associated with. According to participants, communities perceive stroke as a terminal disease and as such may not be willing to spend so much funds in its management but would rather wish that the sufferer dies peacefully. Also, participants stated that stroke occurrence affects economic and productive capacity and overall, the quality of life of both the sufferer and his caregivers. In terms of treatment options, participants mentioned the importance of sufferers in receiving hospital treatment in the event of stroke. However, one barrier that may preclude visit to hospital is the lack of funds to cover associated diagnostic and medical bills. Participants advocated for reduced treatment fees in the treatment and management of stroke in comparison to other health conditions. Also, some participants recommended adoption of unorthodox treatment as another option with the caveat to embrace orthodox treatment approach if unorthodox fails. The statements below support the above findings:

*"He will not be able to take care of his family; his businesses will be affected."; "There is nobody that will know that stroke is about to come, even the person that owns the body cannot know that he will be having the attack. So, it is very certain that one may not be able to curb this affliction"- (FGD Participant, Ibarapa).*

*"It makes one stagnant and also deforms one. More so, it causes trouble for people that are in his surroundings and high spending too. After all, one will definitely spend money. This is what I feared most" – (FGD Participant, Ibadan)*

*"People will be avoiding him because when they realize that he has defecated and urinated on his body. The pains will be too much for the person that has stroke."; "My contribution is that if people should be taken to hospital.... what normally deprive them from going is lack of money. So, I want to say that the money to be paid for this affliction should not be much. For example, if one wants to go for HIV test it is free, so we want them to reduce the money for treatment of stroke sickness so that people could go to the hospital for treatment than for them to go to the traditionalist." – (FGD Participant, Ibarapa).*

3. *Benefits to action*: Participants identified actions taken to prevent stroke, as well as the benefits, barriers, and cues for these actions and self-efficacy based on their own motivation, behavior and their social environment across all groups [44–46]. The different actions include adoption of both hospital and traditional modes of treatment in the face of stroke occurrence. However, a shift to unorthodox treatment became the focus in the event of failed expectations

from orthodox treatment. Other *actions* include community sensitization on risk factor prevention practices such as physical activity and reduction of poor dietary practices among others. On an inquiry about the probability of adoption of such recommended actions, participants indicated that recommendations from their preferred health practitioners are usually adopted. However, some people do not adhere due to issues which revolve around low literacy levels and personality issues. *"If our people, the traditionalists can be accommodated along with the Europeans and research carried out, there will be a way out better than this on those afflicted…If orthodox medicine is joined with our traditional cure, there will be exceedingly great results."; "We must pay attention to the kind of food and drink that are suitable for them. He should not drink certain things, rather he should be doing certain things…he should be engaging in exercise within the surroundings…"; "Some people are very stubborn; also lack of adequate education can cause it. Some people always follow it while some ignore it."- (FGD Participant, Ibarapa)*

4. ***Self-efficacy and Cues to Action***: To explore triggers to adoption of recommended action, participants mentioned the role of education/sensitization on risk factor prevention emphasized practically by the health providers. Other suggested cues to accepting a recommended health action are the positive testimonials and experiences from sufferers. According to them, seeing someone who has had a stroke and had a positive treatment outcome, from either orthodox or unorthodox sources, serves as a positive motivator. With respect to the sources of these health recommendations, participants identified medical specialists as one of those who make recommended health actions that are adopted by community members.

*"We should have lectures … If you don't want this sickness to affect us, the step we can take is for those of us that have hot temper to desist from it. Let's always go for checkup; "...some foods we should not be eating and those that should be more in our food list…. If we have such lectures that this is how we should do things, it will reduce the affliction of this sickness."- (FGD Participant, Ibarapa).*

*"If they observe that someone goes to hospital and is healed, another person will be encouraged to seek similar treatment there." (FGD Participant, Ibadan).*

5. ***Additional recommendations towards stroke prevention***: In looking for ways to prevent stroke, participants mentioned the need for a vaccine which can be given to people to curb sickness that could lead to stroke. They also noted the need for stroke prevention and care education across communities. Further, recommendation included installation of unorthodox medicine practitioners' office on the same site as the hospital where modern medicine providers work to improve stroke care. Additional recommendations, included design of messages in local languages especially to rural dwellers who in some instances, do not have access to sophisticated technologies for receiving health education messages. Participants expressed the need for help (e.g., medical care, medicine, and care support in the home) and wanted more information about how the SISS could help them manage any family member with stroke. Also, a surveillance system should be set up by research programmes such as ARISES in addition to free health care provision for people who are affected by stroke. These were suggestions from the FGD participants in response to the deployment of the stroke informative surveillance system (SISS), an M-health system developed for the timely reporting of stroke events and tracking of burden of stroke in real-time across communities.

*"If this vaccine is taken, people will no longer experience stroke affliction." –* [***FGD Participant, Ibarapa***].

*"Although the statistics indicate a very high level of cell phones in the communities, several persons indicated challenges with this. For example, one person stated that "some people don't have access to phones but if another means of communication can be sourced, maybe on radio, probably local radio, it will make things to be better….Yoruba program is added on radio, it will make it to record further success." – (FGD Participant, Ibarapa).*

*"The help ARISES can do is to set up representatives to be going from house to house in search of people living with stroke affliction in order to render assistance by giving them free health care so that they can live a purposeful life."- (FGD Participant, Ibarapa).*

## Discussion

Our research highlights unique cultural insights elucidated through qualitative research used to explore and contrast community beliefs and practices of health care providers, both unorthodox medicine practitioners, and orthodox medicine practitioners (MDs, PAs, NPs, RNs) as well as those of community leaders and members across the two communities in Nigeria [2]. The major findings illustrate varying ideals and beliefs towards stroke risk factors, prevention strategies, and diagnosis and care for persons with stroke and the burden for their family, as well as the challenges with obtaining care/treatment/medications in their communities, the hospital and in their home.

Findings from this study resonates with those from previous qualitative study conducted across Nigerian and Ghanaian communities to explore knowledge, attitudes, perceptions of stroke risk with a focus on their willingness to participate in a genomics study on stroke [2]. In terms of perception, and based on outcomes such as paralysis and death, participants see stroke as a severe health condition that necessitates immediate intervention. While our participants highlighted free risk factor screening and indigenous community awareness programs, previous studies indicate that some individuals perceive stroke to be avoidable through leading a physically active life, consumption of healthy diets, and strict compliance to recommended drug regimens [2,20].

From the rural participants perspective, community leaders and members shared that most of all community members would first seek care from the unorthodox medicine practitioners, prior to seeking care in the hospital from orthodox medicine practitioners. Unorthodox medicine is age-long practice in Africa [47]. Reasons that support this practice are often related to huge monies needed for the hospital visit and medications versus the lower cost from the unorthodox medicine practitioners. Others include belief that unorthodox medicine practitioners possess ancient/age-long therapies capable of providing rapid recovery, cure from stroke, which is perceived to be more effective when compared with medical interventions. This position aligns with findings from a study conducted in Ghana and Nigeria [47,48]. Similar findings by Chikafu et al. [15] revealed a community whose males opted for several medical approaches, and visits to alternative providers prior to seeking orthodox healthcare facilities in the event of life-threatening health conditions, cardiovascular diseases and preferred treatment by themselves and/or use of herbs because of tradition and manly assumptions [15].

The need for quick recovery especially with ambulation and other functionalities, is a priority goal of a stroke survivor [2]. In a bid to regain independence and economic activities, seeking unorthodox medicine practitioners, who are usually accessible within the community, may become a predominant practice [48,49]. Several unorthodox medicine practitioners shared their interests in working collaboratively with the orthodox medicine providers to improve post-stroke care. This is in line with suggestions by Adekanmbi [48].

The major concern for most people was post stroke care in the home and the effects on the person experiencing a stroke as well as on the family. Most focused on the person who was totally disabled by the stroke---unable to provide self-care and their need for assistance from the family in daily care; thus, a family member had to provide the care. Provision of care is one essential step towards recovery for a stroke patient [50]. It is a major determinant in preventing recurrent events and/or other devastating health outcomes. Ranging from provision of support towards medication adherence, rehabilitation sessions and assistance with mobility, feeding and other routines, these additional responsibilities come with negative outcomes for the carer. Unfortunately, where there are no support systems for the carer in the form of prior training on this new role, assuming this role becomes a daunting task where the carer is left to fix upcoming concerns [51]. Based on our findings, persons with stroke and family members were unable to generate income to meet basic necessities of the stroke survivor and that of his/her family. This is in line with qualitative findings conducted in South Africa [51]. Access to medical, therapies and rehabilitation facilities as part of stroke management requires huge financial involvement. Stroke patients and survivors are limited in their participation in economic and income generation. Involvement of their care givers in stroke care also impacts the financial status and in the long run, quality of life of the family. Hence, carers of stroke survivors require appropriate support interventions to boost their confidence in providing care for their loved one [50].

As we move from the data and results to the theoretical approaches, all questions were centered around the Health Belief Model [30] which proved to be a good model for guiding our data collection. Each theme of the model was addressed in each FGD and KII and the participants had no expressed issues in answering the questions. Although we did not design questions around the Social Ecological Model [19,24], participants identified needs and actions for each level of the model as well as interactions between each level to improve stroke prevention and care. For the individual, many changes in beliefs, attitudes and behaviors were identified but the individual needed support from their interpersonal relationships, especially their family. Organizationally, recommendations included an integrated care model where alternative and traditional healthcare services could be accessed together with collaboration across providers. Changes were needed in the communities---access to healthy food and health care as well as education related to stroke prevention and care in schools, workplaces, and neighborhoods. Actions at this level could focus not only on education but on the physical and social environments by integrating education about stroke into places where people live, learn, work and worship. Perhaps, even developing community gardens where people can have access to healthy foods. At the societal level, the social, cultural, health, economic and social policies could be explored. The data from the KIIs and FGDs suggest the government explore how to improve health care for stroke and make care more accessible at the community level, as well as involving the University of Ibadan and local and regional government in improving stroke prevention and care not only in Nigeria but across Africa. Some of the identified major challenges related to health care systems included inadequate funding (both from families and other sources), limited healthcare infrastructure, and sparsity of health experts (basically neurology and stroke experts) [52]. The above contrasts with findings from a previous study which identified restricted movement from long-term health issues, progressive physiological ache, and apathy [20,53]. There is a concern to delineate rudimentary components of stroke prevention and care to be applied towards greatest impact on health [54] and stroke. For example, Olatunji and colleagues propose using mobile stroke units (MSUs) in mitigating these issues [52] while our team recommends the SISS, a mobile-health community-based interactive electronic surveillance system that can supply first-hand communal figures on exact proportion of stroke [27], correct stroke-related myths/misconceptions, provide correct knowledge on types, signs/symptoms, causal factors, its prevention and discourage delayed presentation to health facility, which is one of the contributors to poor outcomes for acute stroke cases [55].

## Conclusion

Through the use of qualitative methods, concerns relating to the impact of stroke on communities, institutions and service providers have been identified. However, understanding factors related to delivery of optimum stroke prevention and care remain. Additional qualitative and quantitative research studies and collaboration across researchers and systems (community, health and government systems) are needed to design and test methods for improving stroke prevention, early diagnosis of risks that increase stroke, and methods for improving stroke treatment and care. While we note the non-generalizability of these findings to other cultures, there is a need for a synergistic approach by all relevant partners towards harnessing this opportunity to conceptualize innovative research questions, strategic approaches, collation of research findings and multi-site initiatives. The inherent benefits of the above partnership can be expanded through teamwork between clinicians, non-clinicians, health care providers and affiliated systems for the sole purpose of developing new research questions, strategies as well as conduct of meta-analysis to gather on-going studies and research findings across multiple locations. We also need to translate the research into action as we work across systems and countries to improve stroke outcomes, while recognizing the unique features of each. Simply put, we need to work together to develop plans for individuals, relationships, organizations & communities as well as social and economic policies. In other words, it takes all of us to improve stroke prevention and outcomes, and each of us can take small steps. We must explore and create ways to work across individuals and systems to improve stroke prevention, early diagnosis, treatment and care, and explore ways to build family and community support for maximizing recovery and ongoing care for the person with limited recovery.

## Supporting information

**S1 Table. Key informant results from individual interviews [similarities and differences] at the rural (Ibarapa) and urban (Ibadan) sites respectively.**
(DOCX)

**S2 Table. Focus group results from community leaders and members in Ibadan and Ibarapa.**
(DOCX)

## Acknowledgments

We acknowledge and appreciate our team for their role in data collection and transcription across the various sites in each study location: *Olapeju Boladale, Damilola Olawoye-Abiola, Olusola Adewumi Faleye, Yomi Olorunsogbon, Kolawole Amusat Kabiru, Obuene Paul Onyejeboliseh, Oloruntoba Titus Ebo, Imole Ayobami Yemitan.*

## Author contributions

**Conceptualization:** Carolyn Jenkins, Oyedunni S. Arulogun, Bruce Ovbiagele, Mayowa O. Owolabi.

**Data curation:** Ezinne O. Uvere, Carolyn Jenkins, Olayinka J. Adebajo.

**Formal analysis:** Ezinne O. Uvere, Carolyn Jenkins, Ezinne S. Melikam, Oyedunni S. Arulogun.

**Funding acquisition:** Bruce Ovbiagele, Mayowa O. Owolabi.

**Investigation:** Ezinne O. Uvere, Carolyn Jenkins, Oyedunni S. Arulogun, Omotolani T. Ajala, Adewale E. Ayodele.

**Methodology:** Ezinne O. Uvere, Carolyn Jenkins, Oyedunni S. Arulogun.

**Project administration:** Ezinne O. Uvere, Carolyn Jenkins, Bruce Ovbiagele, Mayowa O. Owolabi.

**Resources:** Carolyn Jenkins, Bruce Ovbiagele, Mayowa O. Owolabi.

**Software:** Carolyn Jenkins, Ezinne S. Melikam, Oyedunni S. Arulogun.

**Supervision:** Ezinne O. Uvere, Carolyn Jenkins, Oyedunni S. Arulogun, Mayowa O. Owolabi.

**Validation:** Ezinne O. Uvere, Carolyn Jenkins, Oyedunni S. Arulogun, Omotolani T. Ajala, Adewale E. Ayodele, Michelle Nichols.

**Visualization:** Ezinne O. Uvere, Carolyn Jenkins, Oyedunni S. Arulogun, Omotolani T. Ajala, Adewale E. Ayodele, Olayinka J. Adebajo.

**Writing – original draft:** Ezinne O. Uvere, Carolyn Jenkins, Michelle Nichols.

**Writing – review & editing:** Ezinne O. Uvere, Carolyn Jenkins, Ezinne S. Melikam, Oyedunni S. Arulogun, Omotolani T. Ajala, Adewale E. Ayodele, Olayinka J. Adebajo, Osimhiarherhuo Adeleye, Olumayowa Ogunronbi, Chukwuemeka Nwimo, Michelle Nichols, Oluwafemi A. Popoola, Joshua O. Akinyemi, Raelle Tagge, Rufus O. Akinyemi, Onoja M. Akpa, Ayodeji Salami, Olalekan J. Taiwo, Ayodeji Adebayo, Paul Olowoyo, Temitope Ilori, Olanrewaju Olaniyan, Richard W. Walker, Bruce Ovbiagele, Mayowa O. Owolabi.

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
