## [Decision Letter · Decision Letter 0]

12 Aug 2024

Dear Dr. Owolabi,

Thank you for submitting your manuscript to PLOS ONE. After careful consideration, we feel that it has merit but does not fully meet PLOS ONE’s publication criteria as it currently stands. Therefore, we invite you to submit a revised version of the manuscript that addresses the points raised during the review process.

**ACADEMIC EDITOR:****Request to Editor in Chief: Reviewer 1 has raised concern for plagiarism. I would like the journal to evaluate the article for plagiarism and take appropriate steps.** If no concerns about plagiarism exist, the authors should tackle the reviewer's comments and respond.

We look forward to receiving your revised manuscript.

Kind regards,

Nishant Kumar Mishra, FRCP PhD MD

Academic Editor

PLOS ONE

A clean copy of the edited manuscript (uploaded as the new *manuscript* file)”.

3. PLOS requires an ORCID iD for the corresponding author in Editorial Manager on papers submitted after December 6th, 2016. Please ensure that you have an ORCID iD and that it is validated in Editorial Manager. To do this, go to ‘Update my Information’ (in the upper left-hand corner of the main menu), and click on the Fetch/Validate link next to the ORCID field. This will take you to the ORCID site and allow you to create a new iD or authenticate a pre-existing iD in Editorial Manager. Please see the following video for instructions on linking an ORCID iD to your Editorial Manager account: "https://www.youtube.com/watch?v=_xcclfuvtxQ".

[The study is supported by the National Institutes of Health (NIH) grant: ARISES (R01NS115944‐01).

However, we kindly request for and by this statement, apply for a waiver of the article processing chrages given that this study was conducted in Nigeria.].

5. We note that you have indicated that there are restrictions to data sharing for this study. PLOS only allows data to be available upon request if there are legal or ethical restrictions on sharing data publicly. For more information on unacceptable data access restrictions, please see http://journals.plos.org/plosone/s/data-availability#loc-unacceptable-data-access-restrictions.

6. In the online submission form, you indicated that [Data will be available upon request from the author as data contains recorded voices.].

8.  Concerns have been raised that this manuscript is very closely related to the following papers, of which you are an author:

Jenkins C, Ovbiagele B, Arulogun O, Singh A, Calys-Tagoe B, Akinyemi R, et al. (2018) Knowledge, attitudes and practices related to stroke in Ghana and Nigeria: A SIREN call to action. PLoS ONE 13(11): e0206548. https://doi.org/10.1371/journal.pone.0206548

Our second publication criterion notes that "If a submitted study replicates or is very similar to previous work, authors must provide a sound scientific rationale for the submitted work and clearly reference and discuss the existing literature. Submissions that replicate or are derivative of existing work will likely be rejected if authors do not provide adequate justification." Please see http://www.plosone.org/static/publication.action#results for more information.

Please provide a clear rationale for the necessity of the study in this submission. Please also explain how the work described in this submission differs from and/or advances on that described in the related paper, and which findings can be considered unique to your submitted manuscript.

Thank you for your attention to these requests.

Additional Editor Comments:

Please respond to the reviewer's comments.

Reviewers' comments:

Reviewer's Responses to Questions

**Comments to the Author**

1. Is the manuscript technically sound, and do the data support the conclusions?

Reviewer #1: Yes

Reviewer #2: Yes

2. Has the statistical analysis been performed appropriately and rigorously?

Reviewer #1: N/A

Reviewer #2: Yes

3. Have the authors made all data underlying the findings in their manuscript fully available?

Reviewer #1: Yes

Reviewer #2: Yes

4. Is the manuscript presented in an intelligible fashion and written in standard English?

Reviewer #1: No

Reviewer #2: Yes

Reviewer #1: The authors have conducted a qualitative analysis on the beliefs, attitudes and practices related to stroke prevention, risk factors and modern/ complementary medicine treatment. This is very much a replication of the methods and findings of the earlier SIREN study which the authors have also referenced. In fact, the layout, tables and content are also similar. I hope the plagiarism check has been done by the authors and the journal- If not, I would recommend that.

The title is: Recommendations for Improving Stroke Prevention and Stroke Care. I do not think the paper has and can address this.

Thus, although the concept is not novel and the findings are also similar to the earlier studies, the study could have emphasized on the solutions/ methods which were adopted to improve awareness of stroke and dispel myths and the effectiveness of these methods.

Reviewer #2: Title: Community Voices: Exploring Beliefs, Attitudes, Practices and Recommendations for Improving Stroke Prevention and Stroke Care in Rural and Urban Communities in

Nigeria

Comments:

This is an important qualitative study which explores and contrasts the topic of stroke from alternative and modern medicine perspectives, but at the same time looks at rural and urban perceptions. By doing this, the author is able to comprehensively tackle the topic and bring out all the important issues around beliefs, attitudes, practices and recommendations for improving prevention and care for stroke in a low resource setting.

Unfortunately, the author needs to work on the manuscript to make it a smooth read for publication.

1. The introduction in the abstract and the manuscript should be tailored towards the topic more than constant reference to the research team/group (ARISES)

2. The same applies to the method section which is filled with descriptions about the research team/group and not much detail about recruitment criteria and sampling. Was the sampling method the same in rural vs. urban? alternative vs. modern medicine? The 5 persons who refused to participate due to conflict of time, which group where they from?

3. There are several published works which have touched on one aspect or the other of this research work. Therefore, the author might have to rephrase on this paper being the first to investigate and contrast community beliefs........

4. The results section could have been better reported by focusing on comparisons e.g. rural vs. urban and alternative vs. modern medicine in table form as opposed to tables 3 and 4, which are lengthy and difficult to follow. The recommendations for improving prevention and care for stroke patients could also have been presented in table form.

5. The results section could also have clearly outlined the similarities and differences between rural vs. urban findings with regards to alternative vs. modern medicine for stroke prevention and care.

6. The author should also work on the typographical and grammatical errors in the manuscript.

All in all, this is a good manuscript which makes a significant contribution to the community voice and can be improved for publication.

**Do you want your identity to be public for this peer review?** For information about this choice, including consent withdrawal, please see our Privacy Policy

Reviewer #1: No

Reviewer #2: No

---

## [Author Response · Author response to Decision Letter 1]

16 Dec 2024

1. PLOS requires an ORCID ID for the corresponding author in Editorial Manager on papers submitted after December 6th, 2016. Please ensure that you have an ORCID iD and that it is validated in Editorial Manager. This has been resolved with the recent link of my ORCID number along with the online resubmission platform.

2. In the online submission form, you indicated that [Data will be available upon request from the author as data contains recorded voices.]. All PLOS journals now require all data underlying the findings described in their manuscript to be freely available to other researchers, 1. In a public repository, 2. Within the manuscript itself, or 3. Uploaded as supplementary information. This policy applies to all data except where public deposition would breach compliance with the protocol approved by your research ethics board. If your data cannot be made publicly available for ethical or legal reasons (e.g., public availability would compromise patient privacy), please explain your reasons on resubmission and your exemption request will be escalated for approval.

Thank you for your comment and apologies for lack of clarity in our earlier responses. We have now gone back to check our previous responses obtained by our participants as approved by our Ethics Board. Per the conditions of the informed consent document, the study team destroyed audio recordings after verifying the accuracy of the transcripts. Also, the informed consent signed by participants stated that the study team would employ all means possible to maintain the confidentiality of their personal information and that data collected would be used solely for the purposes of this study, with access limited to authorized study personnel only.

Before we proceed with your manuscript, please address the following prompts: a) If there are ethical or legal restrictions on sharing a de-identified data set, please explain them in detail (e.g., data contain potentially identifying or sensitive patient information, data are owned by a third-party organization, etc.) and who has imposed them (e.g., a Research Ethics Committee or Institutional Review Board, etc.). Please also provide contact information for a data access committee, ethics committee, or other institutional body to which data requests may be sent.

Thank you for your comment and apologies for lack of clarity in our earlier responses. We have now gone back to check our previous responses obtained by our participants as approved by our Ethics Board. Per the conditions of the informed consent document, the study team destroyed audio recordings after verifying the accuracy of the transcripts. Also, the informed consent signed by participants stated that the study team would employ all means possible to maintain the confidentiality of their personal information and that data collected would be used solely for the purposes of this study, with access limited to authorized study personnel only.

b). If there are no restrictions, please upload the minimal anonymized data set necessary to replicate your study findings to a stable, public repository and provide us with the relevant URLs, DOIs, or accession numbers. For a list of recommended repositories, please see

https://journals.plos.org/plosone/s/recommended-repositories. You also have the option of uploading the data as Supporting Information files, but we would recommend depositing data directly to a data repository if possible. We will update your Data Availability statement on your behalf to reflect the information you provide.

Thank you for your comment and apologies for lack of clarity in our earlier responses. We have now gone back to check our previous responses obtained by our participants as approved by our Ethics Board. Per the conditions of the informed consent document, the study team destroyed audio recordings after verifying the accuracy of the transcripts. Also, the informed consent signed by participants stated that the study team would employ all means possible to maintain the confidentiality of their personal information and that data collected would be used solely for the purposes of this study, with access limited to authorized study personnel only.

4. Concerns have been raised that this manuscript is very closely related to the following papers, of which you are an author: Jenkins C, Ovbiagele B, Arulogun O, Singh A, Calys-Tagoe B, Akinyemi R, et al. (2018) Knowledge, attitudes and practices related to stroke in Ghana and Nigeria: A SIREN call to action. PLoS ONE 13(11): e0206548. https://doi.org/10.1371/journal.pone.0206548

Our second publication criterion notes that "If a submitted study replicates or is very similar to previous work, authors must provide a sound scientific rationale for the submitted work and clearly reference and discuss the existing literature. Submissions that replicate or are derivative of existing work will likely be rejected if authors do not provide adequate justification." Please see http://www.plosone.org/static/publication.action#results for more information.

Please provide a clear rationale for the necessity of the study in this submission. Please also explain how the work described in this submission differs from and/or advances on that described in the related paper, and which findings can be considered unique to your submitted manuscript.

Thank you for your comment. Similarities and differences relating to the above studies have been expatiated in the uploaded version of the Response to Reviewers File (attached)

5. Please ensure that you refer to Figure 1 in your text as, if accepted, production will need this reference to link the reader to the figure. This has been addressed accordingly

6. Please change the highlighted text in your manuscript with black text. This has been addressed accordingly

---

## [Decision Letter · Decision Letter 1]

2 May 2025

Dear Dr. Owolabi,

Thank you for submitting your manuscript to PLOS ONE. After careful consideration, we feel that it has merit but does not fully meet PLOS ONE’s publication criteria as it currently stands. Therefore, we invite you to submit a revised version of the manuscript that addresses the points raised during the review process.

We look forward to receiving your revised manuscript.

Kind regards,

Jianhong Zhou

Staff Editor

PLOS ONE

Reviewers' comments:

Reviewer's Responses to Questions

**Comments to the Author**

Reviewer #2: All comments have been addressed

Reviewer #3: (No Response)

Reviewer #4: (No Response)

2. Is the manuscript technically sound, and do the data support the conclusions?

Reviewer #2: Yes

Reviewer #3: Yes

Reviewer #4: Yes

3. Has the statistical analysis been performed appropriately and rigorously?

Reviewer #2: Yes

Reviewer #3: Yes

Reviewer #4: Yes

4. Have the authors made all data underlying the findings in their manuscript fully available?

Reviewer #2: Yes

Reviewer #3: No

Reviewer #4: Yes

5. Is the manuscript presented in an intelligible fashion and written in standard English?

Reviewer #2: Yes

Reviewer #3: Yes

Reviewer #4: Yes

**Reviewer #2:**  The authors have to find a way to streamline tables 3 and 4 which still look untidy, but I am happy with all the corrections made.

**Reviewer #3:** The authors sought to use epidemiological qualitative data to explore non-traditional factors that impact stroke prevention, risks and approaches to improving outcomes. I welcome the "outside the box" thinking to tackle these obstacles. The methodology is sound and rigorous. The paper is well written. References to other studies incorporated. Conclusion is appropriate, except that a statement of non-generalizability of the findings to other cultures should be added.

**Reviewer #4:** This is an excellent manuscript with meaningful findings and discussions. I have a few recommended revisions, with the main revision being an expansion of written results and removal (or substantial shortening) of tables 3 and 4.

Introduction:

- Great work, clear.

Methods:

- Page 6: Can remove the sentence "The COREQ domains, such as research team..." as it is repeating information you've already explained.

- Domain 1 section can be shortened. You provide many details.

- Theoretical models: in your second paragraph on page 10, about the health belief model, you provide clarification in parentheses about what "perceived severity" means. That is helpful and you could provide clarification about each of these factors (perceived susceptibility, etc).

Results:

- Overall, the written section is very short, and you leave it in the hands of the reader to go through all the information in your four tables. I recommend summarizing the key findings (which you partially did within tables 3 and 4). You could include 1 or 2 quotes per theme to illustrate your findings. Tables 3 and 4 should therefore be substantially shortened in order to include some quotation examples. They can even be pushed to supplementary materials if you are able to succinctly synthesize your main findings within the written results.

- Tables 1 and 2 contain identifying information for each participant. To protect anonymity, provide this information in a descriptive general way. For instance, you can indicate mean age and age range, provide education levels in percentages (e.g., "30% of participants have a first degree"), and provide examples of community roles.

- Provide a shorter way to refer to "orthodox/modern medicine/healthcare providers" and "alternative/complementary medicine providers and healers." These terms are lengthy and are used repeatedly throughout, which makes your text hard to read. You can describe these groups and choose one shorter term to refer to them.

- Page 33: at the end of the first paragraph, you can remove the last two sentences (i.e., "It is worthy of mention ..."). These have already been explained your previous text.

Discussion:

- This is the strength of this manuscript. The discussion explores important findings, hypotheses for these results, and future directions. It is comprehensive, clear, and meaningful. Excellent work.

- Page 34: at the end of the first paragraph, you have a grammatical error. You say "previous studies perceive stroke to be avoidable..." but studies do not perceive, people perceive. You can correct this by writing, "previous studies indicate that some individuals perceive stroke to be avoidable..."

Conclusion:

- Strong ending. Great work.

- Page 38: You have a quote but do not cite someone -- please provide a reference for whoever said, "it takes all of us to improve...". Or, if this is not a quote, and the authors are writing this themselves, remove the quotation marks.

Great work overall. Thank you for your important research and for considering my suggested revisions.

**Do you want your identity to be public for this peer review?** For information about this choice, including consent withdrawal, please see our Privacy Policy

Reviewer #2: No

Reviewer #3: **Yes:** Felix E. Chukwudelunzu, MD, MBA, FAHA, FAAN

Reviewer #4: No

---

## [Author Response · Author response to Decision Letter 2]

24 Sep 2025

COMPLETE TABLE SHOWING RESPONSES TO REVIEWERS COMMENTS ARE HIGHLIGHTED IN THE COVER LETTER AND ALSO ATTACHED

Review Comments to the Author (September 24th, 2025)

Please include a copy of Table 1 and 2, which you refer to in your text on page 10.

Thank you for your comment which is well acknowledged. Kindly note that the Tables 1 and 2 referred to under page 10 was a typo error. The participants details mentioned under page 10 have now been presented as descriptives (see previous cover letter and table highlighting all comments below). Hence, appropriate corrections and table numberings have been made on the clean and tracked version of the manuscript.

Review Comments to the Author (July 15th, 2025)

1. Reviewer #2: The authors have to find a way to streamline tables 3 and 4 which still look untidy, but I am happy with all the corrections made.

2. Reviewer #4: This is an excellent manuscript with meaningful findings and discussions. I have a few recommended revisions, with the main revision being an expansion of written results and removal (or substantial shortening) of tables 3 and 4.

Results:

3. Overall, the written section is very short, and you leave it in the hands of the reader to go through all the information in your four tables.

4. I recommend summarizing the key findings (which you partially did within tables 3 and 4). You could include 1 or 2 quotes per theme to illustrate your findings.

5. Tables 3 and 4 should therefore be substantially shortened in order to include some quotation examples. They can even be pushed to supplementary materials if you are able to succinctly synthesize your main findings within the written results. Thank you for your comments.

We have carefully considered reviewer recommendations and further edited tables 3 & 4 and moved them to the supplementary section.

For the results, findings from both tables have been summarised and presented in descriptive format with supporting statements in italics.

6. Reviewer #3: The authors sought to use epidemiological qualitative data to explore non-traditional factors that impact stroke prevention, risks and approaches to improving outcomes. I welcome the "outside the box" thinking to tackle these obstacles. The methodology is sound and rigorous. The paper is well written. References to other studies incorporated. Conclusion is appropriate, except that a statement of non-generalizability of the findings to other cultures should be added. Thank you for your comment. This has been addressed.

Introduction:

• Great work, clear. Thank you for your comment which is well acknowledged.

Methods:

• Page 6: Can remove the sentence "The COREQ domains, such as research team..." as it is repeating information you've already explained.

Thank you for your comment. This has been addressed.

• Domain 1 section can be shortened. You provide many details. Thank you for your comment. This has been addressed.

• Theoretical models: in your second paragraph on page 10, about the health belief model, you provide clarification in parentheses about what "perceived severity" means. That is helpful and you could provide clarification about each of these factors (perceived susceptibility, etc). Thank you for your comment. This has been addressed.

Results

• Tables 1 and 2 contain identifying information for each participant. To protect anonymity, provide this information in a descriptive general way. For instance, you can indicate mean age and age range, provide education levels in percentages (e.g., "30% of participants have a first degree"), and provide examples of community roles.

Thank you for your comment. This has been addressed.

• Provide a shorter way to refer to "orthodox/modern medicine/healthcare providers" and "alternative/complementary medicine providers and healers." These terms are lengthy and are used repeatedly throughout, which makes your text hard to read. You can describe these groups and choose one shorter term to refer to them. Thank you for your comment. This has been addressed. We shorten "orthodox/modern medicine/healthcare providers" with – ‘orthodox medicine’ While we shorten "alternative/complementary medicine providers and healers" with – ‘unorthodox medicine’

• Page 33: at the end of the first paragraph, you can remove the last two sentences (i.e., "It is worthy of mention ..."). These have already been explained your previous text. Thank you for your comment. This has been addressed.

Discussion:

• This is the strength of this manuscript. The discussion explores important findings, hypotheses for these results, and future directions. It is comprehensive, clear, and meaningful. Excellent work.

Thank you for your comment which is well acknowledged.

• Page 34: at the end of the first paragraph, you have a grammatical error. You say "previous studies perceive stroke to be avoidable..." but studies do not perceive, people perceive. You can correct this by writing, "previous studies indicate that some individuals perceive stroke to be avoidable..."

Thank you for your comment. This has been addressed.

Conclusion:

• Strong ending. Great work.

Thank you for your comment which is well acknowledged.

• Page 38: You have a quote but do not cite someone -- please provide a reference for whoever said, "it takes all of us to improve...". Or, if this is not a quote, and the authors are writing this themselves, remove the quotation marks. Thank you for your comment. This has been addressed.

---

## [Decision Letter · Decision Letter 2]

22 Oct 2025

Dear Dr. Owolabi,

We look forward to receiving your revised manuscript.

Kind regards,

Sarah Jose, Ph.D.

Staff Editor

PLOS ONE

Journal Requirements:

Reviewers' comments:

Reviewer's Responses to Questions

**Comments to the Author**

Reviewer #2: All comments have been addressed

Reviewer #4: (No Response)

2. Is the manuscript technically sound, and do the data support the conclusions?

Reviewer #2: Yes

Reviewer #4: Yes

3. Has the statistical analysis been performed appropriately and rigorously?

Reviewer #2: N/A

Reviewer #4: N/A

4. Have the authors made all data underlying the findings in their manuscript fully available?

Reviewer #2: Yes

Reviewer #4: Yes

5. Is the manuscript presented in an intelligible fashion and written in standard English?

Reviewer #2: Yes

Reviewer #4: Yes

Reviewer #2: The authors have addressed all my concerns. By eliminating the tables 3 and 4, an explanation of these findings makes better reading than before. The methods section has also been improved with more detailed explanation of what they set out to achieve. The discussion also flows in tandem with the results.

Reviewer #4: The authors did a nice job of re-working this manuscript, notably expanding on the results and making tables 3&4 into supplementary data. Thank you for addressing the recommended changes.

My main recommendation for revisions is to condense the results section as it is extensively long and remove many of the quotes provided. In qualitative research, a quote should only be provided when it illustrates a point you are explaining. Instead, the authors appear to provide quotes as a way to "back up" their results analysis. This is not needed. Please summarize the main findings and only provide quotes when it helps illustrate a particular point.

Page 16: "bagged a university degree" -- this word is slang. Please reconsider your word choice (e.g., received, obtained, earned).

**Do you want your identity to be public for this peer review?** For information about this choice, including consent withdrawal, please see our Privacy Policy

Reviewer #2: **Yes:** Dr Stanley Zimba

Reviewer #4: No

---

## [Author Response · Author response to Decision Letter 3]

5 Dec 2025

1. My main recommendation for revisions is to condense the results section as it is extensively long and remove many of the quotes provided. In qualitative research, a quote should only be provided when it illustrates a point you are explaining. Instead, the authors appear to provide quotes as a way to "back up" their results analysis. This is not needed. Please summarize the main findings and only provide quotes when it helps illustrate a particular point.

Thank you for your comments.

Overall, the qualitative findings section have been condensed from ten (page 15-25) pages to six (page 15-20) pages. Also supporting statements/quotations have also been revised to retain most relevant ones.

Under the KII findings, all sub-themes under the theme ‘risk susceptibility’ found in pages 14-15, paragraph 1-4 have been condensed into a single paragraph. The sub-themes under the theme, ‘ risk severity’ have been condensed into a paragraph and can be found in page 15-16, paragraph 2. The same approach have been applied to other themes which are now condensed to one paragraph each as found in pages 16– 17.

The same approach was applied to the FGD findings. These findings which occupied 4 pages, have now been condensed to two pages under pages 18-20.

2. Page 16: "bagged a university degree" -- this word is slang. Please reconsider your word choice (e.g., received, obtained, earned).

Thank you for your response. This has been revised in the main document under page 14 paragraphs 2.

---

## [Decision Letter · Decision Letter 3]

2 Jan 2026

Community Voices: Exploring Beliefs, Attitudes, Practices and Recommendations for Improving Stroke Prevention and Stroke Care in Rural and Urban Communities in Nigeria.

PONE-D-24-25604R3

Dear Dr. Owolabi,

We’re pleased to inform you that your manuscript has been judged scientifically suitable for publication and will be formally accepted for publication once it meets all outstanding technical requirements.

Kind regards,

I Gede Juanamasta

Academic Editor

PLOS One

Additional Editor Comments (optional):

Reviewers' comments:

Reviewer's Responses to Questions

**Comments to the Author**

Reviewer #4: All comments have been addressed

2. Is the manuscript technically sound, and do the data support the conclusions?

Reviewer #4: Yes

3. Has the statistical analysis been performed appropriately and rigorously?

Reviewer #4: N/A

4. Have the authors made all data underlying the findings in their manuscript fully available?

Reviewer #4: (No Response)

5. Is the manuscript presented in an intelligible fashion and written in standard English?

Reviewer #4: Yes

Reviewer #4: Thank you for addressing my comments and suggestions. The manuscript appears stronger and clearer now that results have been condensed. It has been a pleasure reviewing your research. This study is important and valuable.

**Do you want your identity to be public for this peer review?** For information about this choice, including consent withdrawal, please see our Privacy Policy

Reviewer #4: No

---

## [Editor Report · Acceptance letter]

PONE-D-24-25604R3

PLOS One

Dear Dr. Owolabi,

I'm pleased to inform you that your manuscript has been deemed suitable for publication in PLOS One. Congratulations! Your manuscript is now being handed over to our production team.

Kind regards,

on behalf of

Dr. I Gede Juanamasta

Academic Editor

PLOS One